# A Non-Linear Convolution Network for Image Processing

**Stefano Marsi [1,\*], Jhilik Bhattacharya [2], Romina Molina [1,3,4] and Giovanni Ramponi [1]**

[1] Image Processing Laboratory (IPL), Engineering and Architecture Department, University of Trieste, 34127 Trieste, Italy; rominasoledad.molina@phd.units.it (R.M.); ramponi@units.it (G.R.)

[2] CSED, Thapar University, Punjab 147004, India; jhilik@thapar.edu

[3] Electronic Department, National University of San Luis (UNSL), D5700HHW San Luis, Argentina

[4] MLAB, The Abdus Salam International Centre for Theoretical Physics, 34100 Trieste, Italy

[\*] Correspondence: marsi@units.it; Tel.: +39-040-5582542

**Abstract:** This paper proposes a new neural network structure for image processing whose convolutional layers, instead of using kernels with fixed coefficients, use space-variant coefficients. The adoption of this strategy allows the system to adapt its behavior according to the spatial characteristics of the input data. This type of layers performs, as we demonstrate, a non-linear transfer function. The features generated by these layers, compared to the ones generated by canonical CNN layers, are more complex and more suitable to fit to the local characteristics of the images. Networks composed by these non-linear layers offer performance comparable with or superior to the ones which use canonical Convolutional Networks, using fewer layers and a significantly lower number of features. Several applications of these newly conceived networks to classical image-processing problems are analyzed. In particular, we consider: Single-Image Super-Resolution (SISR), Edge-Preserving Smoothing (EPS), Noise Removal (NR), and JPEG artifacts removal (JAR).

**Keywords:** neural networks; non-linear convolution; adaptive filters; single-image super-resolution; noise removal; image deblocking; JPEG artifacts removal; edge-preserving smoothing

## 1. Introduction

In digital images, pixels which are spatially close one to each other have highly correlated values. This well-known fact is exploited by image-processing algorithms for image enhancement, noise reduction, interpolation, etc., and to effectively perform image compression and image analysis or understanding tasks. However, it has also been demonstrated [1] that in natural images such an inter-dependency can be better exploited through non-linear functions, rather than through simple convolution-based linear operators.

The same situation holds, in principle, for the class of operators that currently provide the best performance in many of the above applications: Convolutional Neural Networks (CNNs). To realize rich and complex functions most neural network-based systems are composed of a deep stack of linear convolution layers, the outputs of which undergo a point-wise non-linearity, usually a *ReLU*() or a *tanh*().

CNNs using more complicated and ambitious non-linearities (e.g., non-point-wise functions) have of course been proposed too. In most cases, however, these operators are used in a very conservative way since their adoption tends to generate a large increase in the number of parameters of the network and in its training and inference times. In the next section we provide some references about the state of the art in this field.

In the system we propose in this paper,

1.  the linear convolution process used in standard CNNs is replaced by a Non-Linear Convolution (NLC) that more thoroughly exploits the relationships that exist among adjacent image pixels;

2.  the input-dependent weights generated inside the NLC are normalized to preserve

the dynamic range of the data while the input image is transformed into a set of features, processed in the hidden layers, and finally projected into the desired output image;

3.　the NLC modules are trained end-to-end by a standard back-propagation algorithm.

The NLC process that will be described in the following permits continuous and dynamic modification of the weights of the convolution, making the operator *spatially variant* and dependent on the local characteristics of the input image.

The novel contributions provided by this paper can be summarized as follows:

- a new typology of neural network layer, realizing a non-linear convolution is introduced;
- we prove that such a layer performs like a space-variant linear convolution and we suggest how to configure it accordingly to the desired behavior;
- we propose some strategies to combine profitably these layers in larger architectures;
- complete architectures which adopt this layer are proposed and used to solve different image-processing problems.

This paper is organized as follows: in Section 2 we provide some references about recent related research. Then, Sections 3 and 4 present in detail the structure of the proposed non-linear layer and, respectively, discuss its motivations. Section 5 illustrates several possible architectures using the proposed layer, while Section 6 is devoted to a verification of its basic performance in the context of single-image super-resolution (SISR). More detailed application cases are considered in Section 7. Section 8 provides some observations about the realization of the proposed network as an FPGA. Conclusions are drawn in Section 9.

## 2. Non-Linear Convolution Methods in CNN

The literature provides several approaches focused on neural network architectures that exploit non-linear elements different from the canonical activation functions. Some of these methods express the non-linearity in a punctual way, for example by introducing a punctual multiplication operation on the outputs as in [2]. In [3], instead, the authors introduce Volterra kernels in convolutional neural networks, and in [4] a quadratic kernel is introduced too. However, all these methods tend only to provide a better model for the neuron response, following the principle of biologically inspired neural networks. To capture higher-order interactions of features, the kernel pooling [5] is proposed in a parameter-free manner. In [6] Lin et al., propose a "Network in Network (NIN)", as a new method to enrich CNN layers and somehow introduce a non-linearity in the kernel. In [7], an efficient method to apply a learning algorithm to higher-order Boltzmann Machines is proposed. In [8] it is proposed to replace convolution kernels by representing the instantiation parameters as activity vectors via a capsule structure. A deformable convolutional network [9] introduces non-linearity as an affine transform in the learning process. In [10] the non-linear layers called "Modulated Convolutional Network" (MCN) arise from a simplification process, designed to approximate unbinarized convolutional layers while maintaining similar performance and obtaining a considerable reduction of the storage space.

In [11] the authors introduce non-linear effects in the convolution operation in an SVM-like fashion, defining a Gaussian convolution the parameters of which can be trained via gradient descent process. Gaussian convolution layers, standard convolution layers, and possibly other non-linear convolution layer are stacked to form a Hybrid Non-linear (HN) structure. A decisive point is the way in which the components of this stack are selected and ordered. The authors resort to a random search to find the composition that provides the best accuracy.

The non-linearity proposed in [12] aims to take into account efficiently the different perspective views of a 3D scene; accordingly, changing the size and shape of the receptive field. Even if the authors exploit their method for the purposes of scene parsing, their

method looks promising also for image interpolation and geometrical correction, which is per se a still open image-processing problem.

## 3. Single Layer Architecture

### 3.1. The Main Layer

The way in which we develop our new non-linear image-processing system resembles the structure of a series of *N*-size Volterra kernels that operate with fixed weights on tuples of input data $x$ to generate an output $y$, as represented in Equation (1):

$$y(i) = h_0 + \sum_{n=0}^{N} w_n \cdot x(i-n) + \sum_{n=0}^{N} \sum_{m=0}^{N} w_{m,n} \cdot x(i-m)x(i-n) + \ldots \tag{1}$$

Our model, however, differs from the one above for some important properties that permits consideration of the very nature of the data that we analyze and, in particular, of the data we generate: i.e., the pixels of an image.

It must be noted first that if we consider a standard linear convolution, as reported in Equation (2), and we choose to evaluate the weights $w_n$ through further linear convolutions (one for each weight) of the inputs as represented in Equation (3),

$$y(i) = b' + \sum_{n=0}^{N} w_n \cdot x(i-n) \tag{2}$$

$$w_n = b_n'' + \sum_{m=0}^{M} w_{m,n}' \cdot x(i-m), \tag{3}$$

we can substitute Equation (3) in Equation (2) to obtain

$$y(i) = b' + \sum_{n=0}^{N} \left[ b_n'' + \sum_{m=0}^{M} w_{m,n}' \cdot x(i-m) \right] \cdot x(i-n). \tag{4}$$

Now, setting $M = N$ we get the very same expression of a Volterra series limited to the first two kernels. The network we propose uses a non-linear convolution layer similar to the one in (4) as its basic element.

Although in a canonical Convolution Neural Network (CNN) the fundamental element is a convolution layer with fixed weights, the novelty in this network is in using, in the convolution process, a group of normalized, spatial dependent weights $w_n$. Such layers perform as filters able to change their response according to the different parts of the image to be processed. A simplified block scheme of the proposed strategy is depicted in Figure 1.

To describe the behavior of the proposed Non-Linear Convolution Network (NLCN) we start from the simple case of a single input channel and a single output channel.

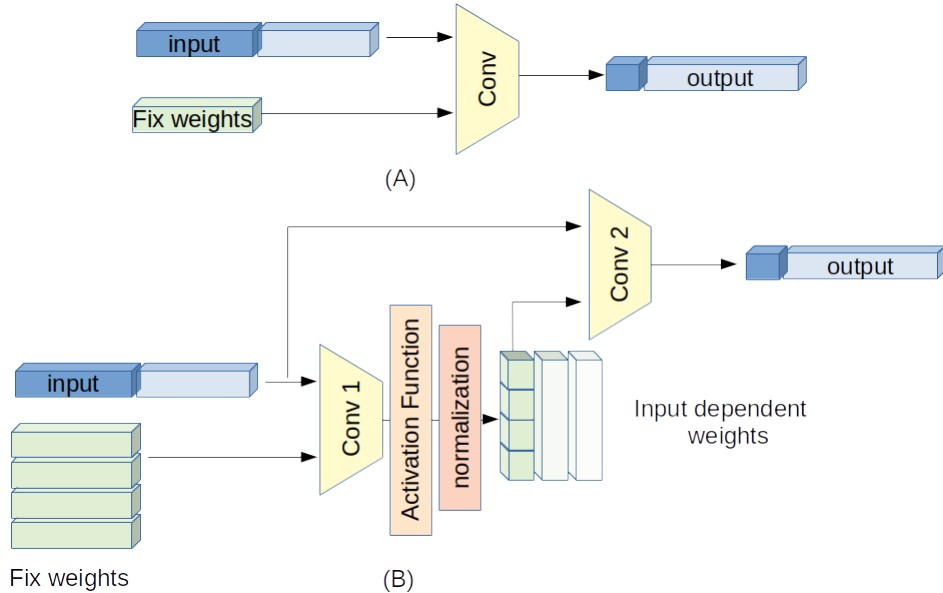

**Figure 1.** A block scheme of the proposed strategy: (**A**) a linear convolution, (**B**) the proposed non-linear (input-dependent) convolution.

*3.2. One-Channel Case*

Consider 2-dimensional input $x(i,j)$ and output data $y(i,j)$. For the sake of simplicity we consider, in the following equations, a square convolution kernel of odd size $W_1 \times W_1$, unitary stride, no dilation and a padding suitable to maintain the output size equal to the input one. The 2D convolution process can be described as in Equation (5).

$$y(i,j) = \sum_{n=1}^{W_1} \sum_{m=1}^{W_1} x(i+n-o_1, j+m-o_1) \cdot v_{i,j}(n,m) \tag{5}$$

where

$$o_1 = \frac{W_1 - 1}{2} \tag{6}$$

Although in a classic linear convolution the weights are fixed and do not depend on the local position of the input samples, i.e., $v_{i,j}(n,m) = v(n,m)$, we introduce a space-variant relationship: every weight $v_{i,j}(n,m)$ of the convolution can vary according to the local characteristics of the input image. To obtain this behavior, the weights $v_{i,j}(n,m)$ are computed through a further convolution of the input data with a kernel of odd size $W_2 \times W_2$ composed by constant weights $u_{n,m}(r,s)$, followed by a non-linear point-wise activation function ($AF$):

$$\hat{v}_{i,j}(n,m) = AF\left( \sum_{r=1}^{W_2} \sum_{s=1}^{W_2} x(i+r-o_2, j+s-o_2) \cdot u_{n,m}(r,s) \right) \tag{7}$$

with

$$o_2 = \frac{W_2 - 1}{2}. \tag{8}$$

The weights undergo a normalization process expressed as

$$v_{i,j}(n,m) = \frac{\hat{v}_{i,j}(n,m)}{\sum_{n,m} \hat{v}_{i,j}(n,m) + \epsilon} \tag{9}$$

or alternatively as

$$v_{i,j}(n,m) = \frac{\hat{v}_{i,j}(n,m)}{\sum_{n,m} |\hat{v}_{i,j}(n,m)| + \epsilon}. \tag{10}$$

It can be noted that combining Equations (5) and (7), setting $W_2 = W_1$ and neglecting the point-wise activation function $AF$ and the normalization process, the final expression corresponds to a second-order Volterra kernel. Furthermore, introducing as commonly done a bias parameter in Equation (7), also the first-order Volterra kernel is represented.

The normalization process is fundamental to guarantee a correct system gain and a controlled dynamic range of the output image. For example, using a ReLU as $AF$ in Equation (7) forces all the weights of Equation (5) to be strictly positive. In such a condition the system expressed by Equation (5) yields a spatially variant low-pass version of the input image. To guarantee that the output image presents a dynamic range comparable to the input one, it is important to force the sum of the weights to be unitary, introducing Equation (9). In such a way Equation (5) operates as a spatial adaptive filter with a unitary DC gain.

There are however other situations where, conversely, it is preferable to carry out a convolution that uses both positive and negative weights. In such a case it is obviously necessary to adopt a different activation function, such as a $tanh()$, that does not limit its output to positive values as a ReLU does. Moreover, the normalization must be adapted to Equation (10) to ensure the output dynamics are constrained within certain limits.

Other different norms and activation functions can also be considered in different cases, according to the task the proposed network should accomplish. Lastly, we include a small $\epsilon$ value in Equations (9) and (10) to avoid the training process to fall into illegal values.

The total number of different weights $u_{n,m}(r,s)$ is actually $W_1^2 W_2^2$ and corresponds to the number of independent parameters that will be configured during the training process.

### 3.3. Multi-Input-Output Channel Case

Now let us consider an input signal $x(i,j,k)$, where k represents the different input channels $1 \leq k \leq K$, and a desired output $y(i,j,l)$ with $1 \leq l \leq L$, where L is the number of desired output channels. In this case, Equation (5) can be rewritten as

$$y(i,j,l) = \sum_{p=1}^{K} \sum_{n=1}^{W_1} \sum_{m=1}^{W_1} x(i + n - o_1, j + m - o_1, p) \cdot v_{i,j,l}(n,m,p). \tag{11}$$

In a classic linear convolution, the weights $v_{i,j,l}(n,m,p)$ do not depend on the local spatial position and could be written as $v_l(n,m,p)$. In the proposed solution, on the contrary, they are variable weights that depend on both the desired output channel $l$ and the local characteristics of the input data surrounding the position $(i,j)$. Like in Equation (7), these weights are determined via a linear convolution layer applied to the input data, using several $W_2 \times W_2 \times K$ kernels (one for each weight), followed by a suitable non-linear point-wise activation function and a normalization process such as the one in Equations (13) and (14).

$$\hat{v}_{i,j,l}(n,m,p) = NL\left( \sum_{q=1}^{K} \sum_{r=1}^{W_2} \sum_{s=1}^{W_2} x(i + r - o_2, j + s - o_2, q) \cdot u_{n,m,p}(r,s,q)_l \right). \tag{12}$$

As with Equations (9) and (10), the normalization process is fundamental to keep the output data within a suitable range and to guarantee a correct gain. The weight normalization can be

$$v_{i,j,l}(n,m,p) = \frac{\hat{v}_{i,j,l}(n,m,p)}{\sum_{n,m,p} \hat{v}_{i,j,l}(n,m,p) + \epsilon} \tag{13}$$

or alternatively

$$v_{i,j,l}(n,m,p) = \frac{\hat{v}_{i,j,l}(n,m,p)}{\sum_{n,m,p} \mid \hat{v}_{i,j,l}(n,m,p) \mid + \epsilon} \tag{14}$$

according to the desired behavior of the system.

The main difference between the single-channel and the multi-channel cases is that while in the former the convolution is performed on a 2-dimensional input domain, in the latter the input domain is 3-dimensional. In the multi-channel case the number of adjustable parameters becomes $W_1^2 W_2^2 K^2 L$.

## 4. Non-Linear Layer Discussion

We analyze in the following the motivations and the strategy for the design of the proposed non-linear convolutional layer. We underline that we limit our discussion to the single layer, rather than to a complete network architecture, for two main reasons:

- usually, a network is developed to deal with a specific application, and is trained on this application. Therefore, comparisons make sense only if limited to that single application. On the opposite, we want to test the *versatility* of the proposed layer;
- the depth of a network, the number of features and the way they are combined, the types of connection, the adoption of processing strategies such as "inception", "residual", etc., all affect its performance. Thus, it is difficult to distinguish the effects of the network architecture from those of the design of the single layer.

Image-processing operators almost always aim to treat differently the different areas which a natural image is composed of (uniform areas, edges, textures), to distinguish noise from detail, natural edges from artifacts, etc. [1]. Researchers often approach the problem using ad-hoc non-linear filters [13] devised through "image analysis strategies" aimed at controlling and tuning the algorithm parameters. Therefore, the non-linearity is not just the extension of a linear model, but it arises from the integration of three separate phases: data analysis, parameters control, and data processing. The double use of input data in the system behavior control and in the actual processing phase leads to (and can be formally described as) the non-linear system.

For example, the authors of [9,12] explicitly follow the three steps of analysis, tuning and processing, actually creating a filter that adapts its behavior according to the different parts of the image by modifying its receptive field through input-dependent spatial transforms, while the frequency response remains constant. This solution is very effective in image classification or semantic segmentation, but would not lead to particular benefits in image-processing tasks. In our algorithm we use the same design strategy but an opposite approach: we maintain the receptive field always centered with respect to the pixel to be processed, and we change the filter frequency response by adapting the filter weights to the various areas of the image. In this way the filter can, for instance, assume a directional behavior when pixels are located on edges or a low-pass behavior in smooth zones.

On the contrary, a more traditional design strategy is pursued by [3], where a convolution layer based on a second-order Volterra kernel is used. The authors use this solution as an extension of the linear case in order to create a system that, with more degrees of freedom, can better emulate the response of a biological neuron. Similarly, in [11], layers with different types of convolutions (linear, Gaussian, L-norm) are alternated. In Section 6 Linear convolutional kernels and Volterra kernels are compared; the latter prove to be superior to the former, but both are outperformed by our operator thanks to its better organization.

It should be emphasized that in general, non-linear networks designed to solve problems of classification or semantic segmentation cannot be plainly ported to solve image-processing problems. Any comparison made across these two different environments should consider the distance that separates them.

## 5. Network Architectures

Like for standard convolutional networks, to realize an operator able to solve real-world problems it is expedient to combine different layers such as the one above described. However, thanks to the much-increased ability in the manipulation of information that the non-linear layer we have devised possesses, the number of such layers can be considerably

smaller compared to the ones used in standard convolutional networks. We have verified, for instance, that using only five layers it is possible to obtain performance comparable to those obtainable by a CNN composed of some dozen layers.

To combine layers, we have tested some strategies, as depicted in Figures 2 and 3:

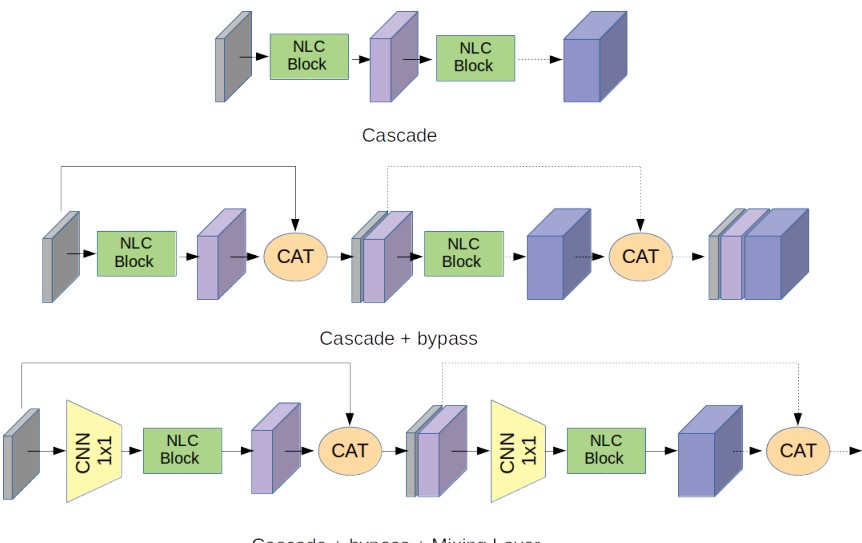

**Figure 2.** Some block schemes to compose architectures.

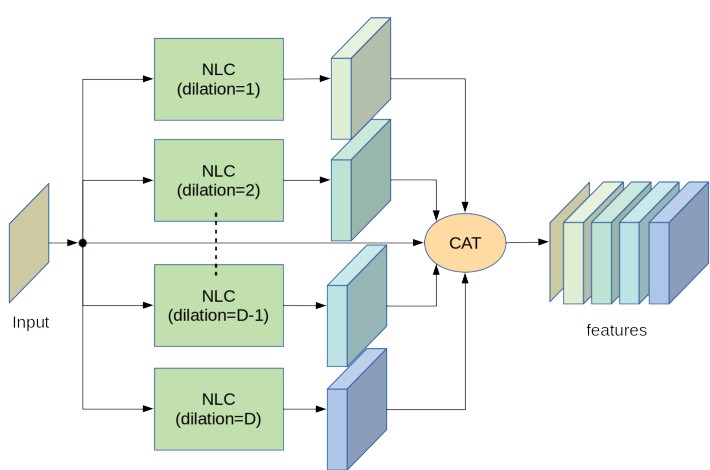

**Figure 3.** A scheme to increase the receptive field.

### 5.1. Cascade

This solution is simple, well-established and adopted by a huge number of CNNs. The architecture is based on a stack of serially connected non-linear layers. In canonical CNNs, the cascade architecture typically suffers from the well-known "vanishing gradient" problem [14]. In our case, however, since the non-linearity of the layer is not a point-wise activation function but is obtained through a quadratic kernel, and since the network can operate using significantly fewer layers, we have verified that this problem is not present.

### 5.2. Cascade + Bypass

To limit the vanishing gradient problem, many authors have introduced short paths in the architecture that bypass portions of the network [14–17], demonstrating significant benefits in both training and inference. Taking a cue from these solutions, direct paths can be introduced in our architecture too.

In our case, these paths are basically used to provide each processing layer with the most complete information available. Each layer, realized with $K$ input channels and $L$ output channels, actually feeds the next layer with $K + L$ overall features. This strategy provides each layer with the richest and the most complete information possible leaving the network choose whether and how to use all this information, during the training.

### 5.3. Cascade + Bypass + Mixing Layer

The previous solution, although quite effective, introduces a significant computational burden. In fact, as the network becomes deeper the number of features to be supplied at each stage increases. Since the network parameters grow with the square of the number of input features, there is a risk of saturating the computing resources soon. To overcome this problem, a further modification that has proved to be effective, is the insertion of a CNN stage with a kernel of spatial dimensions equal to $1 \times 1$ before each NLC layer. This layer operates as a kind of mixer which, taking as input all the input features, feeds the following stage using only suitable linear combinations of them. It should be noted that similar strategies have already been used by other authors, who however adopt simpler operators, such as adders. It is our opinion that in the suggested way the system is much more versatile. In fact, the network during the training phase can choose which features are the most useful, the most significant and what is the best way to combine them. It should also be emphasized that the presence of this very small CNN layer does not significantly increase the computational effort. Moreover, an analysis of the weights the CNN layer assumes at the end of the training can provide information about which features contribute to the final result and in which way.

### 5.4. Parallel Architecture to Increase Receptive Field

Often in image processing it can be useful to analyze large areas of the input images to improve the output effectiveness. This area is called the "receptive field" and there are several techniques to expand it. One method is to increase the spatial dimensions of the convolutive kernel; this solution however increases significantly the number of system parameters, especially in the case of the NLC layer where the number of parameters grows proportionally to the square of the spatial dimensions of both the internal kernels. Another strategy is to stack more and more layers; incidentally, this is the main motivation that led to the birth of deep neural networks. On the other hand, a solution that we have verified to be effective is to adopt a multiscale method: the input data are analyzed by multiple layers operating in parallel. Each one processes input data located in an increasingly large space using an appropriate *dilation parameter*. The results of all these layers are all stacked together with the input data and supplied to subsequent layers. The main scheme is depicted in Figure 3.

## 6. Performance Comparison

To verify that the performances of the proposed NLC system are superior to the ones obtained by canonical convolutional layers we carry out the tests described in the following.

We have identified a simple but not trivial image-processing problem: $2\times$ Single-Image Super-Resolution ($2\times$ SISR). This is a well-established problem that led in the past to the development of many non-linear algorithms, and that recently has contributed to a vast proliferation of dedicated deep neural networks. In this test we do not aim to reach state-of-the-art results: they require very deep and complex architectures. Rather, we adopt this task to compare performances obtainable through different networks of similar complexity, either based on canonical convolutional layers or adopting the proposed ones.

We have therefore developed two very simple non-linear networks, composed respectively of just one or two non-linear layers. A block scheme of the test architecture is shown in Figure 4. In the former case, the network presents only the input layer, realized following the suggested strategy. This layer processes the input image to generate 16 features.

Subsequently, a four-channel $16 \times 1 \times 1$ linear convolution combines with suitable weights these 16 features to form the four final images which contain all the pixels of the final SISR image to be rearranged using the well-known PixelShuffle [18] function. In the latter case the network is slightly deeper: it is composed by two non-linear layers placed in series. The first one generates 4 features while the second one transforms them into 16 features. The last network layer, as in the previous case, recombines these 16 features into the final image. Further details about the networks are reported in Table 1.

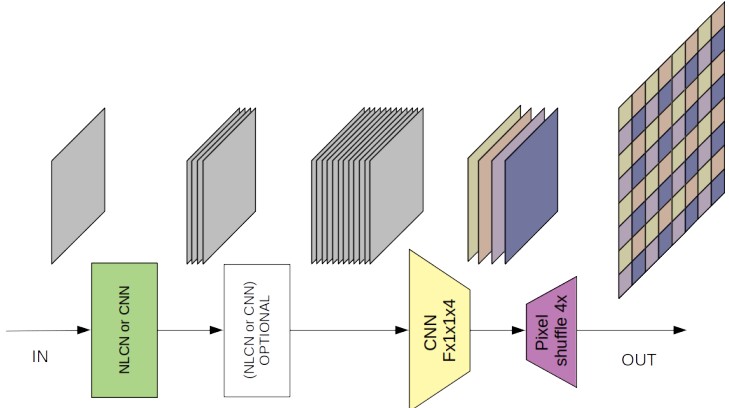

**Figure 4.** A block scheme of the test architecture.

**Table 1.** A comparison among some similar simple networks. Memory (Mem) and Floating-Point Operations (FLOPs) have been estimated for the processing of an $800 \times 600$ image.

| Net Name | Type | Layers | Architecture | Features | Param. | Mem | FLOPs |
|---|---|---|---|---|---|---|---|
| LINNet S | Linear | 2 | 1-16-4 | 16 | 144 | 8 M | 69 M |
| HUGENet S | Linear | 2 | 1-128-4 | 128 | 1152 | 61 M | 550 M |
| NLCN S | Non-linear | 2 | 1-16-4 | 16 | 1296 | 8 M | 626 M |
| VoltNet S | Non-linear | 2 | 1-16-4 | 16 | 1296 | 8 M | 626 M |
| LINNet M | Linear | 3 | 1-4-16-4 | 16 | 612 | 10 M | 293 M |
| HUGENet M | Linear | 3 | 1-16-128-4 | 128 | 18,576 | 64 M | 8.9 G |
| NLCN M | Non-linear | 3 | 1-4-16-4 | 16 | 21,060 | 10 M | 10 G |
| VoltNet M | Non-linear | 3 | 1-4-16-4 | 16 | 21,060 | 10 G | 10 G |

To compare equitably our NLCNs with CNNs, two pairs of similar architectures have been created, where all the non-linear layers have been substituted by canonical convolutive layers. In detail, for each NON-linear architecture, two different CNNs have been built, formed by the same number of layers but with a different number of hidden features: in the former case the features generated by each layer is equal to those generated by the corresponding NLCN, while in the latter the number of features is made significantly larger providing that the number of configuration parameters of the NLCN and of the CNN should be quite the same.

Moreover, to compare the performance of the proposed structure with a different kind of non-linear operator too, another pair of networks based on a combination of a linear and a quadratic Volterra kernel [3] has been realized. These networks have been designed with the same number of layers, parameters and hidden features as the NLCNs, to make them very similar to the proposed architecture; however, they do not adopt the normalization process introduced in Section 3.

The reasons to limit the comparison to very simple networks which obviously are very far from solving the SISR problem with performance close to the state of the art is two-fold. First, we wanted to focus the performance evaluation only on the single layer, without being influenced by the context. In fact, one can define different strategies to connect layers

realizing very creative architectures, but doing so it is difficult to determine whether the benefit is due to the structure of the layer or to the architecture in which it is employed. Secondly, it should be considered that the sub-sampled image used as input for the SISR algorithm has actually lost a part of its information that no method will ever be able to recover completely. This means that all methods must deal with an asymptote in their performance. The better the network, the closer its performance to such asymptote, and consequently the less evident the differences between various methods. A proof of this is that a dozen algorithms developed in the last two years present very similar performance, in a range of less than 1 dB in PSNR [19]. Staying far from this limit allows us to make the comparison between the networks more evident.

All the networks have been trained on the same dataset composed by 2500 different gray level images. At each step, a random crop of 256 × 256 pixels has been chosen as target for the network output, while its 0.5× bicubic decimated version is adopted as input data. At the end of each training epoch the networks have been tested using the luminance data of "Set5", a very common benchmark for this sort of problems.

The result of the proposed test is depicted in Figure 5. We can notice that the NLCN architecture outperforms the CNN not only when the number of features employed are the same, but also when they are significantly larger. Moreover, also a very simple NLCN such as "NLCN-small" can outperform a CNN network such as "HUGENET-medium" which needs a larger number of both parameters and features, is deeper and can even take advantage of a larger receptive field. Moreover, it can be noted that the proposed architecture, when adopted for an image-processing task, also outperforms networks based on Volterra kernels [3]. Notwithstanding Volterra-based networks exploit very general-purpose linear and quadratic convolution kernel; they do not introduce any strategy to control the compatibility of the output signal dynamics. This makes the strategy adopted in our architecture very promising for image-processing problems, compared to other commonly used linear and non-linear convolutional layers.

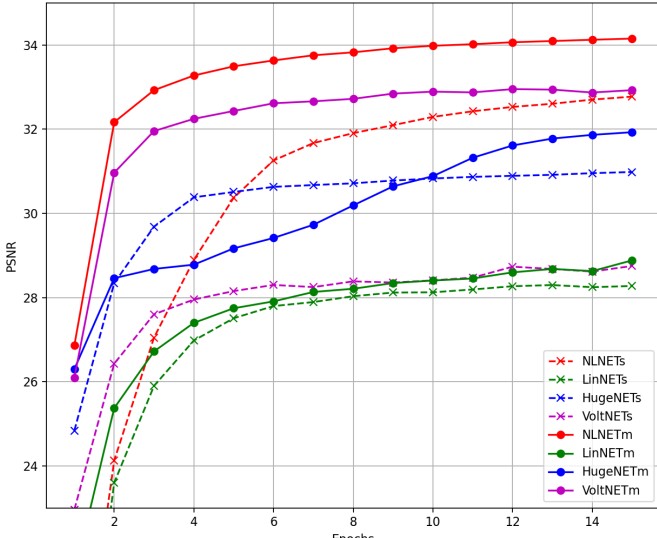

**Figure 5.** *Cont.*

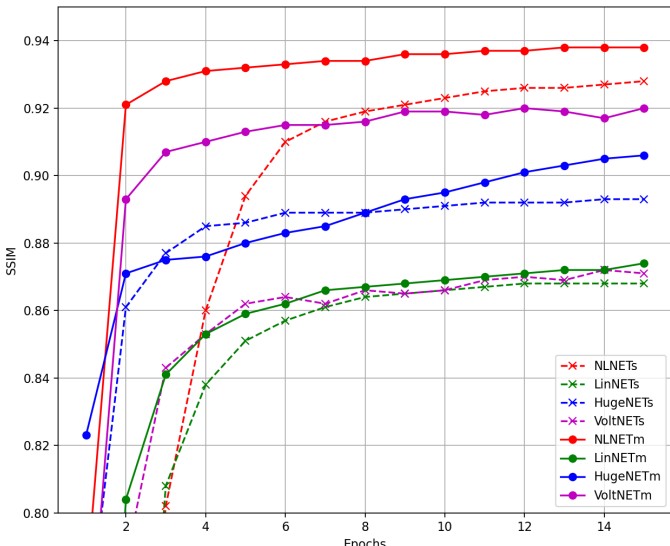

**Figure 5.** Networks Training Comparison.

## 7. Network Applications

To verify that the proposed structure is suitable also for creating deeper networks, we integrated it into more complex systems to be applied to different image-processing problems. In particular, we have considered three different applications: Edge-Preserving Smoothing (EPS), Noise Removal (NR), and JPEG artifacts removal (JAR).

### 7.1. Edge-Preserving Smoothing Architecture

The first task we consider is Edge-Preserving Smoothing (EPS), i.e., the ability of a system to remove high frequencies from an image keeping the edges of the objects as sharp as possible. Several algorithms are present in the literature to address this task, and recently also deep neural networks have been used [20–22], none of them however adopt non-linear convolutional kernels.

The network used for this problem is depicted in Figure 6. It can be noted that it adopts the blocks that we have already described in Section 5: An input stage (IAS) for a multiscale input analysis which adopts 4 NLC blocks working in parallel with different dilation parameter respectively $\{1, 3, 5, 7\}$, followed by a series of eight Non-Linear Feature Generator (NLFG) blocks each one of which generates 8 output features. In the end all the features from all the previous stages are recomposed to the output using a CNN with a kernel with a $1 \times 1$ spatial dimensions that actually provides the input with a suitable linear combination of all the features.

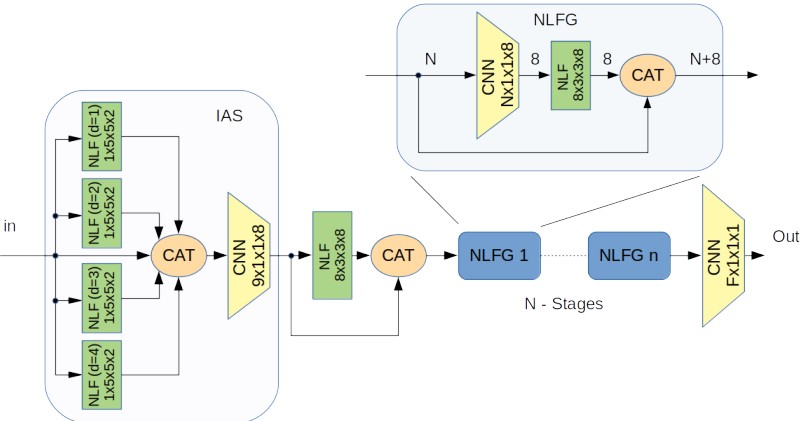

**Figure 6.** A block scheme of the Net architecture.

Since there is no public database able to provide a sufficiently large number of EPS examples to be used during the training phase, we have built a synthetic one. It consists of a theoretically infinite set of randomly generated synthetic images that present both sharp edges and uniform areas. On each of them an additional image is superposed, composed by only high frequency details randomly extracted from a database of suitably processed real-world images. The network has been trained, using the superposition of the two images as the input and forcing the output to reconstruct only the first one, removing as much as possible the information content of the second. An example of these images is shown in Figure 7.

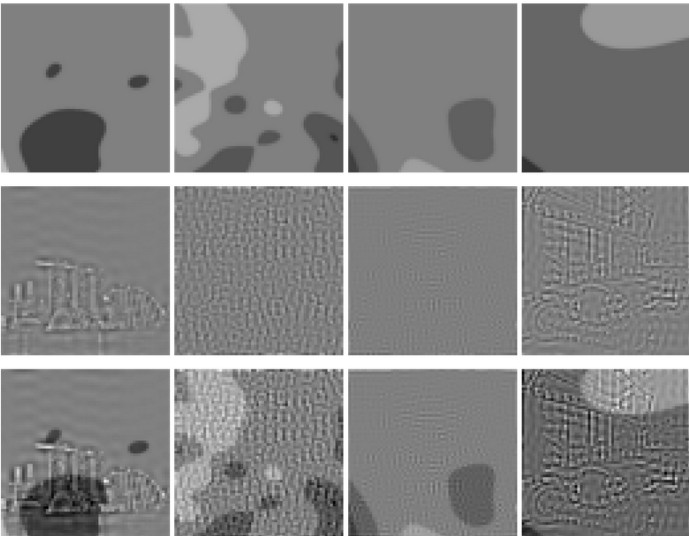

**Figure 7.** Samples of the synthetic images used to train the EPS Network.

At the end of the training, which required about 50 epochs each one consisting of 300 randomly generated items, the network succeeded in carrying out its task. An example of inference obtainable from the network is shown in Figure 8.

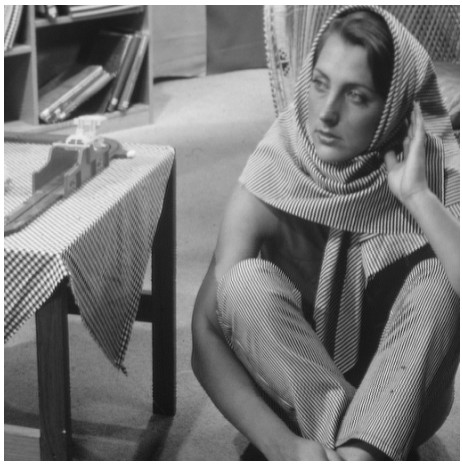
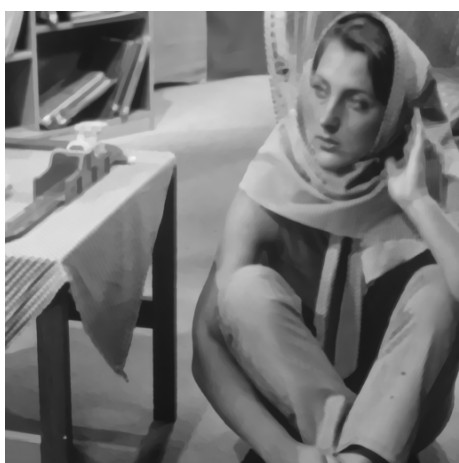

(**a**) Input image                    (**b**) Output Image

**Figure 8.** *Cont.*

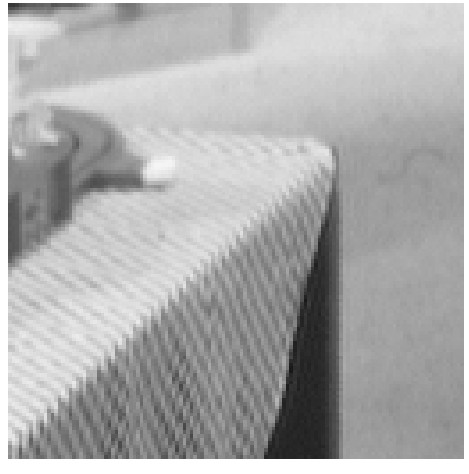 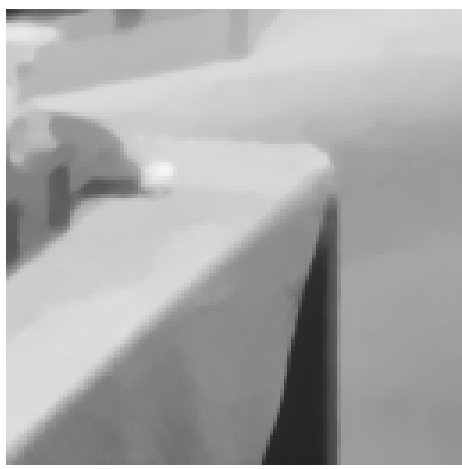

(**c**) Input image crop                 (**d**) Output image crop

**Figure 8.** Application of the Edge-Preserving Smoothing, trained Net.

*7.2. NLCN Performance Showcase for Noise Removal*

As mentioned previously, we want to test the performance of the proposed network on yet another common but important image-processing task, i.e., noise reduction. We target for this purpose natural scene images as well as medical X-ray images. Noise reduction from natural scenes without the risk of oversmoothing fine details is a challenging task, and it becomes even more crucial in case of medical X-ray images. For natural images, we use the BSD-100 dataset for training and the LIVE dataset for testing; for X-ray images, the Chest X-ray dataset collected by Indiana University is used [23].

The network used for this problem makes use of a 2-column architecture, the output of which is concatenated in the end. One column basically provides an output based on the combination of input data, weighted by positive coefficients. This works as a non-linear low-pass filter. The other column adopts a different normalization function, hence providing both positive and negative coefficients. This is particularly suitable for realizing high-pass non-linear filters. We have used Poisson noise, Gaussian noise and a combination of Poisson and Gaussian noise for verifying the results on the X-ray images. Only Poisson noise on both grayscale and color images are used for natural scene images. We have used the popular BM3D filter, as well as a couple of CNNs as a baseline to compare our results both visually and using objective scores via SSIM and PSNR. Such metrics have been adopted taking into consideration that they are widely used in the literature as a standard measurement metrics for these tasks. It should be noted that non-linear convolution-based kernels have not yet been applied for noise reduction tasks, to the best of our knowledge. We have taken FFDNet [24] and GROUP-SC (G-SC) [25] as NN bases to compare performances on the X-ray and LIVE datasets. The parameter settings for the NLCN are the same for all experiments. Both the NLCN and FFDNet have a faster inference time (less than one second per image) when compared to BM3D and Group-SC. Group-SC has the highest average inference time of approximately 77 s per image for color images and 34 s per image for grayscale images. However, the number of trainable parameters for Group-SC are lower (approx. 112 K) than both FFDNet (553 K) and NLCN (414 K). It is observed that the NLCN obtains overall competitive results. It provides a superior performance on X-ray images when compared to the other NNs. Performance of these networks on Gaussian noise is particularly low. An example of these images is shown in Figures 9 and 10. It can be observed that the BM3D filter and the NNs do a very fine job in removing noise from the natural as well as X-ray images. However, for BM3D, in both cases the images are over-smoothed. The FFdNet and Group-SC work better on natural images, but oversmooth the X-ray images. This is also reflected in the SSIM and PSNR scores shown in Table 2. It should be noted that the main purpose of these

experiments was to demonstrate that the network is suitable for a wide range of tasks. Particular performance enhancement and robust comparison with several state-of-the-art techniques is not a part of our scope and can be obtained by changing different parameters such as kernel sizes, number of filters etc.

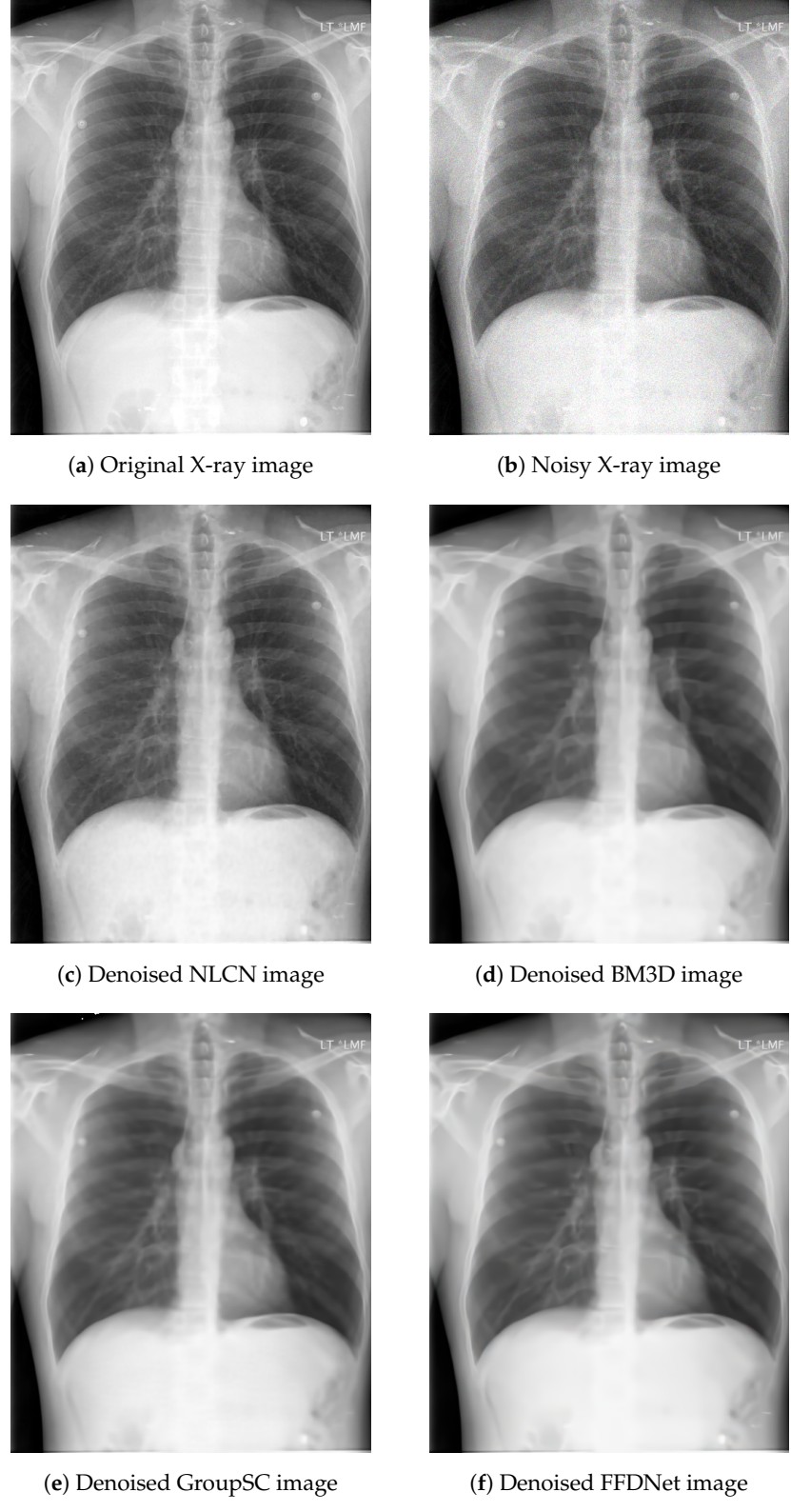

(**a**) Original X-ray image          (**b**) Noisy X-ray image

(**c**) Denoised NLCN image          (**d**) Denoised BM3D image

(**e**) Denoised GroupSC image          (**f**) Denoised FFDNet image

**Figure 9.** Application of denoising on X-ray images.

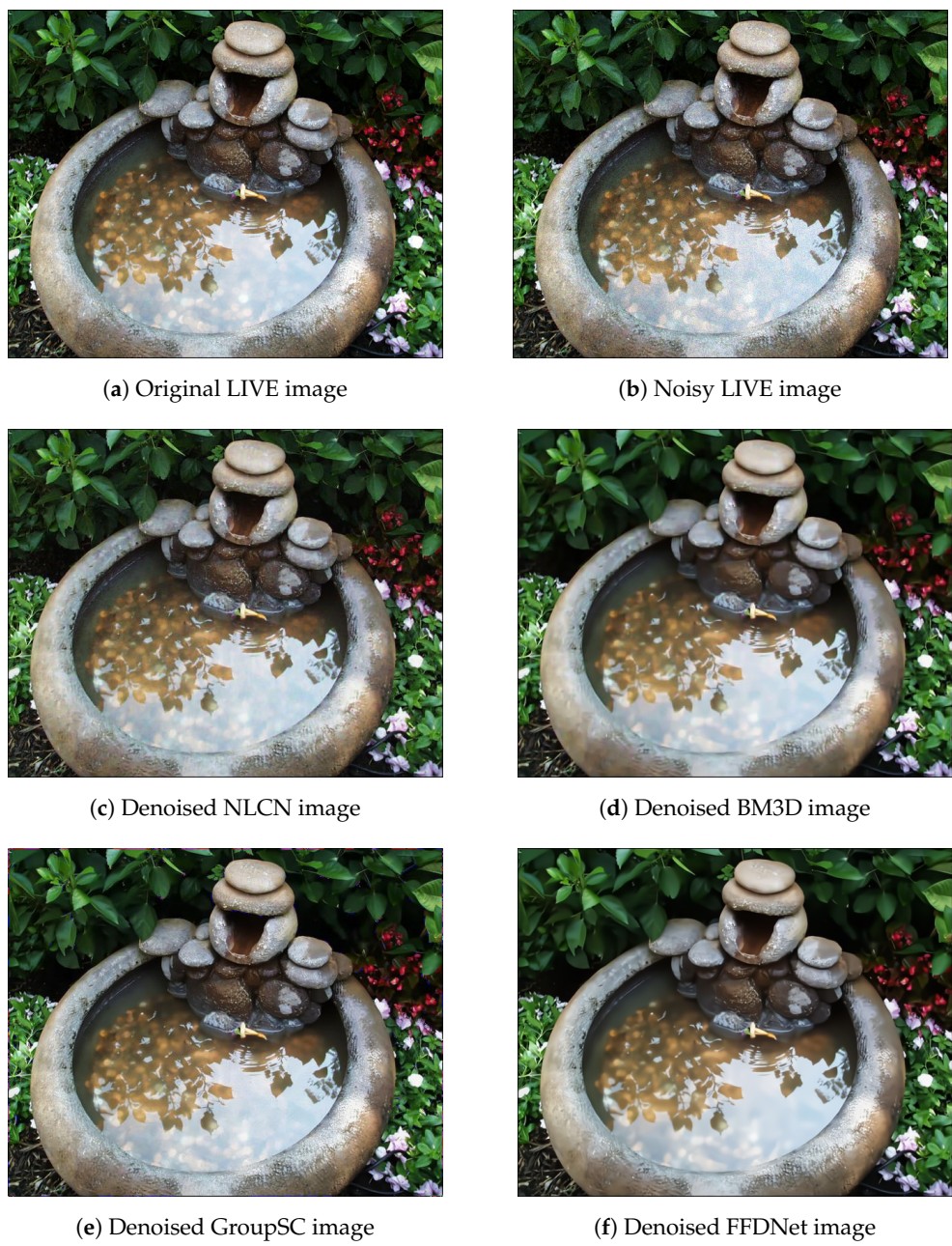

(**a**) Original LIVE image

(**b**) Noisy LIVE image

(**c**) Denoised NLCN image

(**d**) Denoised BM3D image

(**e**) Denoised GroupSC image

(**f**) Denoised FFDNet image

**Figure 10.** Application of denoising on LIVE dataset.

**Table 2.** SSIM and PSNR metrics with denoised images from gray(g) and color(c) LIVE(L) and X-ray dataset(x).

| Type | SSIM | | | | |
| --- | --- | --- | --- | --- | --- |
| | Noisy | NLCN | BM3D | FFD | G-SC |
| Pois_x | 0.672 | 0.961 | 0.948 | 0.947 | 0.940 |
| Gaus_x | 0.169 | 0.895 | 0.852 | 0.858 | 0.396 |
| Pois+Gaus_x | 0.520 | 0.950 | 0.688 | 0.942 | 0.916 |
| Pois_L_g | 0.701 | 0.899 | 0.853 | 0.907 | 0.870 |
| Pois_L_c | 0.789 | 0.883 | 0.837 | 0.893 | 0.894 |
| **Type** | **PSNR** | | | | |
| | Noisy | NLCN | BM3D | FFD | G-SC |
| Pois_x | 31.00 | 41.47 | 40.01 | 39.99 | 39.52 |
| Gaus_x | 20.97 | 36.56 | 33.03 | 33.16 | 29.12 |
| Pois+Gaus_x | 27.38 | 39.86 | 21.76 | 38.83 | 39.52 |
| Pois_L_g | 27.61 | 33.52 | 30.43 | 32.94 | 32.93 |
| Pois_L_c | 30.06 | 32.82 | 29.77 | 32.35 | 33.00 |

*7.3. NLCN Performance Showcase for JPEG Artifacts Removal*

The last application we consider for the NLCN is JPEG artifacts removal. In this case, due to JPEG compression, ringing and blocking effects are seen on the images. We train the NLCN to remove these artifacts from the images. We use the BSD-100 dataset for training and LIVE dataset for testing. For all cases we use a quality factor $Q = 20$. We demonstrate visual and objective scores via SSIM and PSNR. An example of these images is shown in Figure 11 and Table 3. We have used the DnCNN [26] tool, available on MATLAB, to compare our results. It can be observed that our filter does a better job compared to DnCNN. The network used for this problem is the same as the one used for noise reduction with the exception that it uses dilations in the convolution layers. We have used both grayscale as well as color images to verify our results. As mentioned earlier, these experiments were mainly designed to demonstrate that the network is suitable for a wide range of tasks. Further performance enhancement may be obtained by changing different parameters such as kernel sizes, number of filters etc. However, in general, the ability to have spatially dependent, variable filter responses for a non-linear convolution network proves to be more beneficial for de-blocking and de-ringing the image, when compared to traditional linear convolution networks. This allows a local texture-based reconstruction of the image without oversmoothing. This can be observed from the processed image in Figure 11. In case of the DnCNN version an oversmoothing effect is observed while removing the blocks. The same is not the case for the NLCN image.

**Table 3.** SSIM and PSNR metrics with images from LIVE dataset after JPEG artifacts removal.

| Type | SSIM | | | PSNR | | |
| --- | --- | --- | --- | --- | --- | --- |
| | JPEG | NLCN | DnCNN | JPEG | NLCN | DnCNN |
| LIVE_gray | 0.860 | 0.867 | 0.869 | 29.24 | 32.80 | 29.76 |
| LIVE_color | 0.860 | 0.865 | 0.869 | 29.30 | 32.64 | 29.76 |

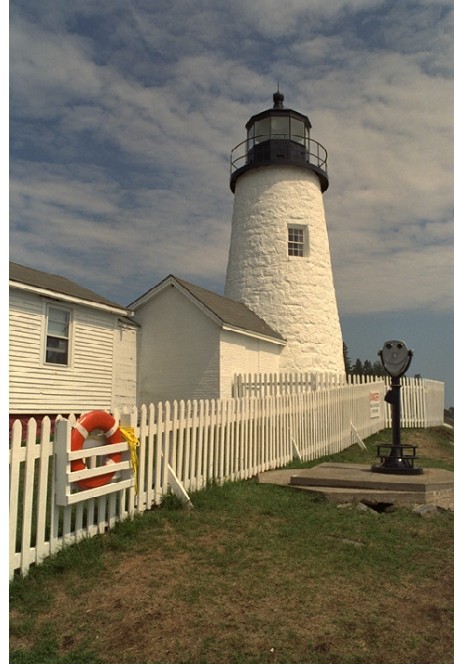

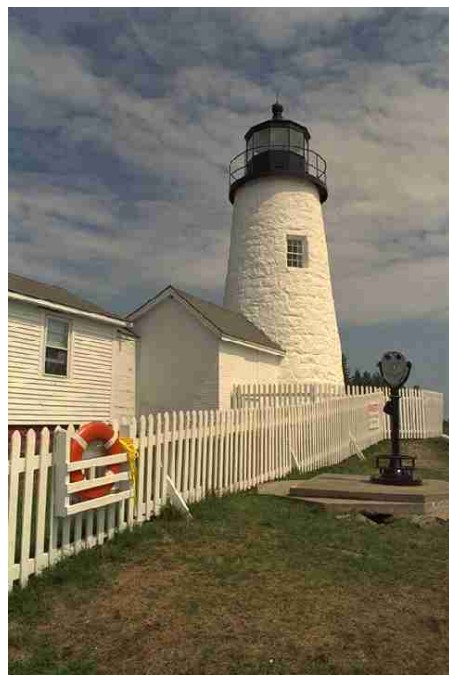

(**a**) Original LIVE image        (**b**) LIVE image with JPEG artifacts.

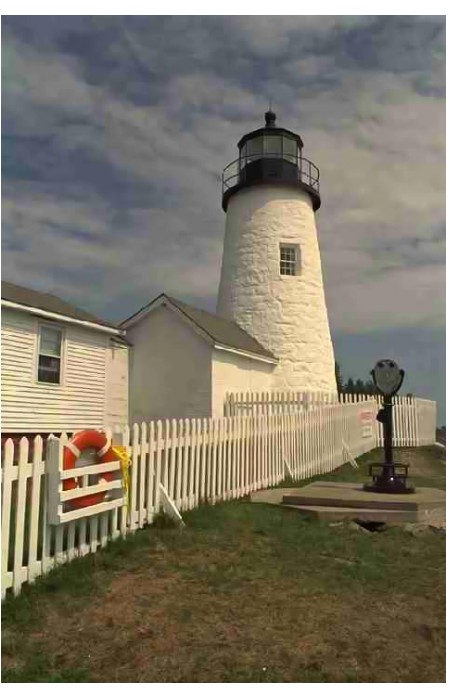

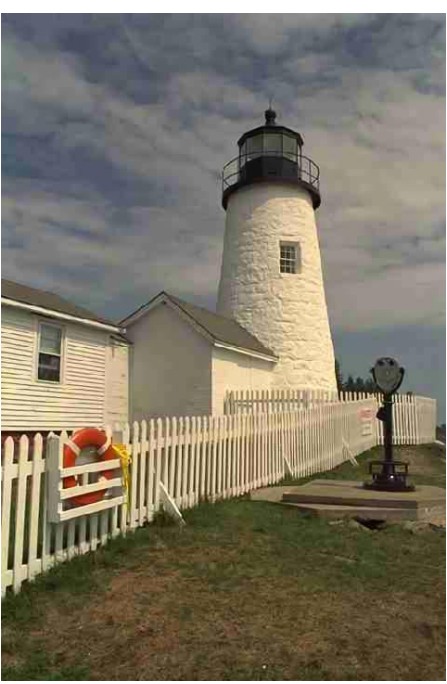

(**c**) Artifact removed NLCN image        (**d**) Artifact removed DnCNN image

**Figure 11.** Application of JPEG artifact removal on LIVE dataset.

## 8. Perspectives for an FPGA Realization of the NLCN

In recent years, hardware accelerators have become more popular to implement neural network inference tasks. Examples such as mNet2FPGA [27], hls4ml [28], NEURAghe [29], Angel-Eye [30], among others, support the rapid growth of framework development for the inference task in different fields of research. A detailed review in this topic is presented in [31].

Presently, field programmable gate arrays (FPGAs) are the main platform to accelerate machine learning inference in the edge and in the cloud, because of their features such

as reduced power consumption, high parallelism, low latency, dynamic reconfiguration, availability of dedicated digital signal processors (DSPs), among others.

In FPGA technology, compression techniques are suitable to reduce redundant parameters and memory footprint, which has direct impact in the power consumption, speed and resource use [32–35]. Cheng et al. [36] presented a review of the state of the art in compression techniques, summarizing the different approaches in: parameter pruning and quantization, low-rank factorization, transferred/compact convolutional filters and knowledge distillation. The first category is the most used in the field of hardware accelerators because it is related with reducing the precision of the weights and bias; on the other side, it allows the reduction of the model removing redundant neurons and connections, which has a direct impact in the number of DSPs used by arithmetical operations.

In an embedded machine learning approach the algorithm will be directly mapped into RTL level and metrics such as energy consumption, accuracy, throughput/latency should be considered, to have an optimal system balanced in computation and memory transaction.

The proposed NLCN is based on fewer layers than canonical CNNs for the cases of study presented in this paper, allowing a significantly lower number of features, which is ideal in embedded accelerators due their limited resources. Deep CNNs, on the other side, are demanding on memory and computing.

The memory usage in the NLCN inference is dominated by the memory to store the parameters, the memory to save intermediate data, and the space for internal computation. On the other side, computation is mainly affected by multiplication and addition operations in the different layers.

With FPGAs, operational intensity (or computation intensity) is the ratio among the number of operations (arithmetical, indexing and comparison) that are performed, and memory traffic is defined as the total amount of data transferred between memory and FPGA device (which implies the number of read and write operations and the data type). The amount and type of operations implied with NLCN will be affected by the final hardware deployment strategy.

Considering Equations (5) and (7) for a single channel, with $W1^2$ and $W2^2$ as kernel sizes, the I feature maps at the input and the O feature maps as output, the number of parameters for one layer is defined in Equation (15):

$$L\_parameters_{NLCN} = W1^2 \times W2^2 \times I^2 \times O \tag{15}$$

Although for the CNN the number of parameters is defined in Equation (16), for one layer with kernel size $W1 \times W1$, and with a bias term for each feature map.

$$L\_parameters_{CNN} = ((W1^2 \times I) + 1) \times O \tag{16}$$

Comparing these equations, we can observe how the feature maps at the input and the second kernel size W2 affect the number of parameters in the NCLN layer, which is directly related with memory usage.

Regarding the final size of the model, it can be obtained by the relation between data structures generated to contain weights and bias and their data-type size (32-floating point, 16-fixed point, 8-fixed point, among others); that is, the product between the total amount of parameters and the data-type size. In the context of FPGA technology, the data type specified could be different for each layer, which will have a direct impact in the final size of the model, since each layer will be multiplied by its required bit-width.

In the context of image processing, a drawback is the size of the number of pixels transferred from off-chip memory to the FPGA on-chip memory to perform the inference task, which can lead to a memory-bound system, so several strategies should be implemented to obtain an optimal compromise between computation and off-chip memory transaction. The NLCN is a good candidate to perform the inference task reducing the data transferred to the FPGA part. Nevertheless, the operators in the activation functions should

be improved for this type of accelerators, to reduce the impact of complex mathematical operators.

In NLCN architecture, each layer has its input to perform its own processing composed by all the features developed by the previous layers. In FPGAs, this behavior should be addressed to avoid computational and bandwidth bounds due the amount of data generated between the different layers, strategies such as loop unrolling and pipelining, memory partitioning can be used to optimize performance and bandwidth requirements. Different approaches for the implementation of CNNs in FPGA have been proposed in the literature, aiming to exploit the different levels of parallelisms: the feature-map-level, operator-level, task-level and layer-level [37–40], among others, and the same considerations should be taken into account in the context of NLCNs to have a fully and optimized implementation in FPGA technology.

The implementation of NLCNs into FPGA leads to the opening of a new spectrum of applications in embedded systems, as the final model obtained requires less memory and computation, with a great potential to improve energy efficiency and speed-up of the inference task. Also, if necessary, the weight analysis presented in this paper can be used to perform the pruning stage for model compression.

## 9. Conclusions

We have introduced in this paper a new type of convolutional neural network for image processing, featuring non-linear convolution layers that permit the learning of spatially variant operations, in this way reducing the network complexity necessary to solve a specific problem. We have demonstrated through several examples that architectures based on the proposed layer can be trained and operate successfully in different processing tasks. We also briefly commented on the advantages that our NLCN offers in terms of its FPGA realization.

We hope this contribution can have a seminal role for a series of studies that will permit the reaching of better compromises between the effectiveness and the computational requirements of CNNs.

**Author Contributions:** Conceptualization, S.M.; methodology, S.M., J.B., R.M., G.R.; software, S.M., J.B.; formal analysis, S.M.; investigation, S.M., J.B., R.M., G.R.; writing—original draft preparation, S.M., J.B., R.M., G.R.; writing—review and editing, S.M., G.R.; funding acquisition, G.R.. All authors have read and agreed to the published version of the manuscript.

**Funding:** This research received no external funding.

**Acknowledgments:** S.M., R.M. and G.R. gratefully acknowledge the support of the University of Trieste.

**Conflicts of Interest:** The authors declare no conflict of interest. The funders had no role in the design of the study; in the collection, analyses, or interpretation of data; in the writing of the manuscript, or in the decision to publish the results.

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
