# Peer review of "A Non-Linear Convolution Network for Image Processing"

_electronics, doi:10.3390/electronics10020201_

Round 1

Reviewer 1 Report

The topic seems to be good, and the presentation is fine enough. But, as a reader, I am not convinced yet how the input dependent convolution kernels can be selected. If it is possible, then it could mean that training data are used accordingly to the test data. To me, it is a start of overfitting. 

Author Response

We are grateful to Rev. 1 for his comments. In the Introduction of our modified paper, we further clarify that the input-dependent convolution kernels are determined via the backpropagation algorithm. In all our experiments we keep the training and testing data totally separated, as necessary to avoid overfitting.
We have improved the methods’ description in Sec.4, the conclusions, and in particular the presentation of the results (See Fig5, Tab1 and Tab2), according to the evaluation marks provided by Rev. 1.

All the relevant (and non-trivial) changes made to our submission have been made evident using a red text font in the new manuscript.

Reviewer 2 Report

The authors present an interesting approach to the domain of neural networks. However, they establish that a non-linear convolution network is a novelty, and they compare their approach with convolutional networks. Regarding that point, I found similar references. Why do not the authors propose comparisons with them? What is the motivation to not include these references and other similar ones in the manuscript? I think the related work/background must be improved reflecting the real novelty of the approach. Finally, the experiments should include some non-linear approach to see what is the best one. Summarizing, the authors must demonstrate that their approach is novel, good, and have a nice performance in comparison with other similar.

Next, possible references:

Zhang, X., Wei, K., Kang, X., & Li, J. (2020). Hybrid nonlinear convolution filters for image recognition. Applied Intelligence, 1-11.

Lei, T., Barzilay, R., & Jaakkola, T. (2015). Molding cnns for text: non-linear, non-consecutive convolutions. arXiv preprint arXiv:1508.04112.

Author Response

We sincerely thank Rev.2 for his careful reading of our contribution, and for providing constructive criticisms and suggestions.

In our modified paper, we extend the description of the panorama of methods available to introduce nonlinear effects in the convolution. For this purpose, we moved a portion of the Introduction to an independent dedicated Sec. 2 (Non-linear convolution methods in CNN).

Later, in another new Sec. 4 (Non-linear Layer discussion), we refer to Sec. 2 comparing the method we propose to the existing literature, and improving the methods’ description.

In particular, we thank Rev.2 for pointing out the paper [Zhang20_Hybrid], very recently published, which we had missed. In our modified paper we provide a description of the approach used there. Instead, we would drop the paper [Lei15_Molding], since it is devoted to the very different problem of text processing. Given its complexity, it is far from obvious to port the same model to image processing applications.

As required, in Sec. 6 we add a comparison with another non-linear convolution technique (See Fig.5 ad Tab1) and in Sec. 7 we add some experimental comparison with other methods based on neural networks. Sec.7 also provides more details about the computational requirements of our method, with respect to other available techniques

All the relevant (and non-trivial) changes made to our submission have been made evident using a red text font in the new manuscript.

Reviewer 3 Report

Dear authors,

        you presented to use space-variant coefficients for convolution to build non-linear-convolution networks. You have demonstrated the effectiveness of the proposed network for multiple image processing tasks. We have the following suggestions:

  1. We suggest that in the Introduction, the difference to existing methods like deformable convolution should be highlighted and discussed.
  2. We suggest that you create a figure to illustrate the proposed non-linear layer in Section 2.
  3. What is the meaning of M and N? They should be clarified. Other symbols should also be clearly introduced.
  4. Most of the results are qualitative results. We suggest that the authors also demonstrate the effectiveness of the proposed network for image classification or semantic segmentation tasks by presenting the accuracy of the network compared to traditional CNNs.
  5. In addition to the parameters, the computation complexity and the memory requirement of the proposed network can also be analyzed.
  6. Some related works can be mentioned and discussed: [*] "Perspective-adaptive convolutions for scene parsing." IEEE transactions on pattern analysis and machine intelligence 42.4 (2019): 909-924. [*] "Acnet: Attention based network to exploit complementary features for rgbd semantic segmentation." 2019 IEEE International Conference on Image Processing (ICIP). IEEE, 2019.

For these reasons, a major revision is suggested.

Sincerely,

Author Response

We sincerely thank Rev. 3 for his careful reading of our contribution, and for providing constructive criticisms and suggestions.

In our modified paper, we extend the description of the panorama of methods available to introduce nonlinear effects in the convolution. For this purpose, we moved a portion of the Introduction to an independent dedicated Sec. 2 (Non-linear convolution methods in CNN).

Later, in another new Sec. 4 (Non-linear Layer discussion), we refer to Sec. 2 comparing the method we propose to the existing literature.

In particular, we thank Rev. 3 for pointing out [Zhang20_PerspectiveAdapt], which we had missed. In our modified paper we provide a description of the approach used there (see Lines 181-191). The nonlinearity introduced there aims to take into account efficiently the different perspective views of a 3D scene, accordingly changing the size and shape of the receptive field. Even if the Authors exploit their method for the purposes of scene parsing, their method looks promising also for image interpolation and geometrical correction, which is per se a still open image processing problem.

Instead, we would drop the paper [Hu19_ACNET], since it is devoted to the very different problem of the segmentation of RGB+depth images. The architecture proposed in that paper, moreover, is based on ResNets and on Attention Complementary Models, which operate using standard convolution components.

About a figure in Sec. 2: As the Reviewer kindly suggested we have added a block scheme of the proposed strategy in Fig. 1.

About M and N: we have changed the indices in Eq. 1 to make more clear the relationship between the proposed structure and Volterra series kernels.

In the modified paper, and especially in Sec. 4, we clarify that the primary purpose of the network we propose are image processing applications like smoothing, deblocking, noise removal, interpolation. Image analysis tasks are presently outside the scope of our contribution. However, we think that architectures based on our nonlinear convolution layer can be devised that will provide satisfactory results also in image segmentation and recognition. We plan to devote future work to this goal.

We did not limit to a qualitative subjective analysis; we included quantitative metrics that are generally used when evaluating image processing tools: the Peak Signal-to-Noise Ratio (PSNR) and the Structural Similarity Index (SSIM). We provide these values for all the applications we consider: Sec. 6 provides PSNR and SSIM values for an image interpolation problem (2xSISR); Sec. 7 shows again PSNR and SSIM for both graylevel (medical) and color images, and for denoising and deblocking problems. The only application in which it is not possible to objectively measure the quality of a processed image is stylization (edge-preserving smoothing), a couple of examples of which are shown in Fig. 7 as an effect of edge-preserving smoothing. We have modified our paper to briefly discuss this situation.

Sec. 7 also provides more details about the computational requirements of our method, with respect to other available techniques

All the relevant (and non-trivial) changes made to our submission have been made evident using a red text font in the new manuscript.

Round 2

Reviewer 1 Report

I think the modified version removes the anxiety of overfitting.  

Author Response

We would like to thank the reviewers for their valuable suggestions.

In this final version

- We romoved the "Highlights" from all senteces
- We emproved resolution In Fig. 5
- We modified Tab.1 to consider FLOPs operations.

Reviewer 2 Report

The manuscript has been upgraded successfully.

Author Response

(The authors gave the same response as above.)

Reviewer 3 Report

Dear authors,

        most of the concerns have been addressed. We suggest that this paper be accepted after solving the following minor issue:

       Mem (memory) and Mops should be spelled out when they are first introduced in the manuscript. What are mops? How about FLOPs and Parameters, which are widely used metrics to measure the complexity of the network?

Sincerely,

Author Response

(The authors gave the same response as above.)
